# Population Health Status of the Republic of Kazakhstan: Trends and Implications for Public Health Policy

**DOI:** 10.3390/ijerph182212235

**Published:** 2021-11-22

**Authors:** Gabriel Gulis, Altyn Aringazina, Zhamilya Sangilbayeva, Zhan Kalel, Evelyne de Leeuw, John P. Allegrante

**Affiliations:** 1Unit for Health Promotion Research, University of Southern Denmark, Degnevej 14, 6700 Esbjerg, Denmark; ggulis@health.sdu.dk; 2Caspian International School of Medicine, Caspian University, 521 Seifullin Street, Almaty 050000, Kazakhstan; 3Kazakh Scientific Research Institute of Eye Diseases, 95a Tole Bi Street, Almaty 050000, Kazakhstan; j.sangilbaeva@gmail.com; 4Kazakhstan School of Public Health, 19a Utepov Street, Almaty 050060, Kazakhstan; halelzhan@mail.ru; 5Centre for Health Equity Training, Research and Evaluation (CHETRE), University of New South Wales, South Western Sydney Local Health District Population Health, and Ingham Institute for Applied Medical Research, Sydney, NSW 2052, Australia; e.deleeuw@unsw.edu.au; 6Department of Health and Behavior Studies, Teachers College, Columbia University, 525 West 120th Street, New York, NY 10027, USA; jpa1@tc.columbia.edu; 7Department of Sociomedical Sciences, Mailman School of Public Health, Columbia University, 722 West 168th Street, New York, NY 10032, USA

**Keywords:** global health, Kazakhstan, life expectancy, population health, public health policy, social determinants of health

## Abstract

The Republic of Kazakhstan began undergoing a political, economic, and social transition after 1991. Population health was declared an important element and was backed with a substantial commitment by the central government to health policy. We examine key trends in the population health status of the Republic of Kazakhstan and seek to understand them in relation to the ongoing political, economic, and social changes in society and its aspirations in health policy. We used the Global Burden of Disease database and toolkit to extract and analyze country-specific descriptive data for the Republic of Kazakhstan to assess life expectancy, child mortality, leading causes of mortality, disability-adjusted life years, and causes and number of years lived with disability. Life expectancy declined from 1990 to 1996 but has subsequently recovered. Ischemic heart disease, stroke, and chronic obstructive pulmonary disease remain among the leading causes of death; child mortality for children under 5 years has declined; and cardiovascular risk factors account for the greatest cause of disability. Considering its socioeconomic development over the last two decades, Kazakhstan continues to lag behind OECD countries on leading health indictors despite substantial investments in public health policy. We identify seven strategic priorities to improve the efficiency and effectiveness of the health care system.

## 1. Introduction

The Republic of Kazakhstan (further Kazakhstan) placed itself on the global public health map with the Primary Health Care Declaration of Alma Alta in 1978 [1], even before it gained independence from the former Union of Soviet Socialist Republics in 1991. The principles of *Primary Health Care* were recently reinforced in the *Astana Declaration* in October 2018 [2]. With a population of approximately 18.5 million people living in an area of 2.73 million square kilometers (km^2^), Kazakhstan is the largest land-locked country of the world, and it is one of the most sparsely populated [3]. Following the collapse of the Union of Soviet Socialist Republics (USSR) in 1991, Kazakhstan entered a political, economic, and social transition, with the oil and gas sector being the principal driver of its rapid economic growth.

Today, Kazakhstan is a country with a relatively young population and a dynamic demographic profile. Following a decline in the population between 1992 and 2002, the population has grown by 20%. The Gross Domestic Product (GDP) was estimated to be USD 180.2 billion in 2019, with a per capita gross national income of USD 8810 compared to USD 40,115 in other OECD countries [4]. Nevertheless, as an oil-exporting country, Kazakhstan has taken a leading place among other Central Asian countries.

Since its independence and as its economy has grown, Kazakhstan has placed an increasing amount of political attention on population health status and public health services. In order to improve the availability and efficiency of health services as well as to ensure equal access to health care, Kazakhstan’s central, regional, and municipal governments have embarked on a phased reform of the health care system, starting with the State Program for Reforming and Development of Healthcare of the Republic of Kazakhstan, which took place from 2005–2010. This reform preceded the “Salamatty Kazakhstan” State Program from 2011–2015 [5], the “Densaulyk” State Program from 2016–2020 [6], and the current State Program for the Development of Healthcare of the Republic of Kazakhstan, which is planned to take place between 2020–2025 [7]. As a result of the new investments in health care, today, the entire population of the country has the right to access to basic social benefits and an expanded primary health care system for free. In addition, the hospital sector has been restructured to reduce dependence on inpatient care [5].

Improving population health is complicated by the unique geospatial position of Kazakhstan. First, although Kazakhstan is the 9th largest country in the world and 4th largest in Asia, it is among the most sparsely populated [8]. The average density is slightly less than 6.93 people per km^2^ (184th place in world population density). Second, two of the world’s largest environmentally hazardous sites are located in Kazakhstan: the Baikonur Cosmodrome and the Semipalatinsk nuclear test site. Following the collapse of the USSR, the cosmodrome became the property of Kazakhstan and was leased by the Russian Federation. Biosphere pollution caused by hazardous rocket fuel constitutes a considerable threat to the local environment, and the Semipalatinsk nuclear test site has increased the risk of cardiovascular disease and cancer in residents living near its location [9].

Given the geopolitical significance of Kazakhstan to the Central Asian region, this paper examines the trends in the health status of the population of the country and discusses their implications for the health care sector and for the reform of public health policy and practice in Kazakhstan and the surrounding region. 

## 2. Materials and Methods

### 2.1. Data Sources

We used the Global Burden of Disease (GBD) database and toolkit [10] to extract and analyze country-specific data for Kazakhstan in order to compare it to other countries in the GBD database. The database and toolkit are in the public domain and are freely available.

### 2.2. Measures

We examined time trends of life expectancy, leading causes of mortality, leading causes of disability adjusted life years (DALY’s), risk of premature mortality due to non-communicable diseases, infant mortality, and maternal mortality as key indicators to describe population health trends. Conceptual definitions of these terms are available [11]. To observe changes in health care access and quality, we used the health care access and quality index developed by Fullmann [12]; data for Kazakhstan were extracted from the annex of that paper [12].

### 2.3. Statistical Analyses

We used descriptive analysis only and mostly focused on the presentation and discussion of a time series of indicator data that we believe provide a good representation of the current status of population health.

## 3. Results

### 3.1. Life Expectancy

Life expectancy at birth declined from 1990 to 1996 but started to recover after 1996 (Figure 1). 

A substantial gender gap, however, has persisted, with males having a significantly shorter life expectancy than females. On average, life expectancy for the urban population is only a few months higher than it is in rural areas.

### 3.2. Child Mortality

Mortality for both under-5 year and under-1-year live births has declined steadily since the mid-1990s. Over the past decade, there has been a substantial decline in the mortality rate of children under 5 years of age. During this period, 3744 lives of children under the age of 5 were saved, of which almost 70% of cases (2427 cases) were caused by diseases in the perinatal state, i.e., during pregnancy and in the first week after childbirth. Since 2009, 575 deaths from respiratory complications (mainly from pneumonia) have been avoided [13]. Mortality resulting from congenital malformations is also decreasing; a total of 520 lives have been saved, which is a 47% decrease in mortality from these causes since 2009. Congenital malformations are closely related to the lifestyle of the parents, especially the mother; adequate prevention and monitoring during pregnancy is also beneficial [9]. Thus, from 2009 to 2018, the mortality rate among children under the age of 5 has been reduced by almost half. This is largely due to improvements in obstetric services and better care for preterm births, for the treatment of severe infectious diseases in children (such as pneumonia, diarrhea, sepsis in newborns), and for the treatment of acute malnutrition [14].

### 3.3. Leading Causes of Death

From 1990 to 2019, cardiovascular disease and cancer were the two leading causes of mortality (Figure 2).

Respiratory infections and TB dropped from 3rd place in 1990 to 7th place in 2019; in contrast, digestive organ diseases climbed up from 9th in 1990 to 3rd in 2019. Ischemic heart disease and stroke remain the most frequent causes of death in 2019, followed by chronic obstructive pulmonary disease (COPD). Compared to 2009, diabetes has now become one of the 10 most important causes of death. Over the past two decades, infant and maternal mortality has declined four- and six-fold, respectively, which has allowed Kazakhstan to approach the OECD average. Kazakhstan also has low mortality from infectious and parasitic diseases, but the prevalence of tuberculosis remains an alarming exception. Nevertheless, most deaths are due to chronic conditions, and mortality rates remain much higher than those in OECD countries [15]. However, compared to the other countries of the Central Asian region, Kazakhstan’s indicators are average, although Mongolia and Uzbekistan show significantly higher mortality rates because of NCDs.

### 3.4. Disability

Among the most important causes of disability from 1990 to 2019 are mental health, musculoskeletal disorders, non-communicable diseases, neurological disorders, and unintentional injuries (Figure 3). 

### 3.5. Health Care Access and Quality

Although the health care access and quality indices show a slight deterioration between 1990 and 2000, significant improvement occurred after 2000. Since 2020, there has been an active increase in effort to implement compulsory social health insurance, which supports better access and health coverage while simultaneously maintaining the guaranteed volume of medical care that is provided free of charge. The health insurance package includes consultative and diagnostic assistance, including laboratory services, outpatient drug provision in excess of the guaranteed volume of medical care, inpatient care (except for cases of treatment within the guaranteed volume of medical care), and medical rehabilitation treatment. The guaranteed volume of medical care will be valid for all citizens and foreigners who are permanently residing in the country, regardless of their status in the insurance system. 

The operational mechanism of primary health care was developed with the support of the WHO and was approved in November 2020 by 194 states of the world at the 73rd session of the World Health Assembly. This document is an implementation plan for the Astana Declaration on Primary Health Care (PHC) and offers countries several strategic and operational levers to strengthen PHC [16]. In 2019, 747 hospital organizations and 3204 outpatient clinics were registered in Kazakhstan. The volume of services provided in the field of health care and the provision of social services in Kazakhstan measured in the cost of services provided by healthcare organizations in the amount of funds that come from enterprises, organizations, and (or) directly from the population by the Bureau of National Statistics Agency for Strategic Planning and Reforms continues to grow.

The state guarantees equal access for all categories of citizens to medical services at the primary level, emergency and urgent medical care, and diagnosis and treatment of socially significant diseases (tuberculosis, HIV, mental and behavioral disorders, malignant neoplasms) as well as major chronic non-infectious diseases [17]. In 2019, 14,992.2 thousand people were insured, and the coverage of the compulsory health insurance system was 80.5% [16]. Since then, Kazakhstan has introduced Unified Aggregate Payment (UAP), which provides a simplified procedure for registering the activities of informal workers with the tax authorities [18]. Due to informal workers being outside of the legal framework, the aforementioned payment plan was introduced to allow them access to the health insurance.

## 4. Discussion

Since gaining independence in 1991, Kazakhstan has made impressive economic progress. In addition, numerous strategic reforms have been undertaken to expand access to health services, modernize their delivery, and reduce dependence on hospital care, along with considerable investment in public health infrastructure [19,20,21,22]. Despite reforms and new policies that are intended to support improved population health, Kazakhstan’s health care system continues to face many challenges—many of which are associated with the unique geospatial, geopolitical, and economic realities that hamper innovation—and its leading indicators of health continue to lag behind other OECD countries. These include a lack of modern information technology systems, limited accountability, and a lack of effort to develop effective prevention services and programs to reduce the current trends in the leading causes of mortality.

The sparse and dispersed rural population and geographical features also create challenges for delivering effective medical care across the country. In an effort to strengthen health care delivery in rural areas, a telemedicine network was introduced in 2004. The strategic plan of the Ministry of Health for 2017–2021 calls for the full implementation of the network by 2050, anticipating that digital and mHealth technologies and other “smart services” will achieve improvements in prevention, diagnosis, treatment, and management in rural areas.

In many ways, the organizational structure of Kazakhstan’s health care system today is equal to the structure of most OECD countries. Although the reforms that have been enacted during the last decade are numerous and ambitious, much less attention has been directed toward the implementation and scaling of these reforms at all levels of the health care system and in all corners of the nation.

The best hope for implementing and evaluating reforms in healthcare are state programs. The Ministry of Health and Republican Center for Healthcare Development work in conjunction to monitor the success of the current program and discuss the outcomes each quarter. On the basis of the state programs, modest reforms are being developed to achieve set goals. The latest report on the 2020–2025 state program focuses on three main directions: (1) formation of adherence to a healthy lifestyle among the population, (2) life and development of public health services, and (3) improving the quality of medical care and sustainable development of the health care system; toward these ends, 31 of 111 planned activities have been completed. Although some of the goals have been achieved in some areas, a sharp increase in maternal mortality was recorded during the summer of 2020 as a consequence of the COVID-19 (SARS-CoV-2) pandemic, accounting for 63% of deaths from the total number of cases in the country [23]. Accordingly, in the case of maternal mortality, causes not related to pregnancy, childbirth, and the postpartum period (the so-called indirect causes) increased from 58% (2019) to 72% (2020) [23]. In February 2020, the Center for Analysis of Monitoring of Socio-Economic Reforms was created by the order of the president, but reports have yet to be published [24]. Moreover, the reliability of the data on the level and quality of implementation of these reforms remains a concern, and evaluation of the impact of ongoing changes on improving intermediate and longer-term health outcomes is rarely undertaken.

Following a period of decline in the 1990s, life expectancy increased through 2015 [25]. The decline in child mortality both among children under 5 and under 1 years of age is evidence of the impact of the substantial changes that have been made in maternal and child health care. Despite the investments in health care that have been among the priorities of domestic policy during the last decade, they have led to only modest health status improvements. The trend for men could be related to lower mortality due to external causes, such as fewer road traffic crashes in rural areas; the trend for women may reflect the lower quality of obstetric and gynecological care available in rural areas compared to in urban areas. There are several reasons for this.

First, public expenditures on health care account for only 1.8% of GDP, which covers only 58% of total expenditures in this area, thus leading to significant costs for patients who have to pay for medical services out of pocket. Out-of-pocket payments for patients account for approximately 38% of total healthcare expenditures in Kazakhstan, which exceeds the norm of 20% established by WHO. A large portion of this cost is due to the fact that insurance covers a limited number of drugs, whereas in OECD countries, households pay approximately 40% of the cost of drugs out of pocket on average; in Kazakhstan, this number is 84% [15]. The drugs prescribed by primary healthcare doctors are usually paid for by the patient, with drugs only being provided free of charge to patients with “socially significant diseases”, as defined by the Order of the Minister of Health of the Republic of Kazakhstan, 23 September, 2020, No. RK MoH-108/2020.

Second, a key challenge for any health care system is to protect individuals from significant and/or unexpected health care costs and to ensure that people do not forgo basic care for financial reasons. Achieving this insurance indemnification function requires both broad coverage (e.g., coverage across the entire population, as in the case of Kazakhstan) and low out-of-pocket payments.

Third, although the modernization of the health care workforce in Kazakhstan is underway, serious problems remain with its territorial distribution. Moreover, despite the large number of medical schools and other related institutions [26], only one institute—the Kazakhstan School of Public Health—prepares professionals for work in population health, and it continues to be chronically underfunded. Efforts to prepare and expand the workforce for practice in the primary health care sector are of particular importance because the number of general practitioners per 1000 population is 0.28, which lags behind the OECD average of 0.72; there is a similar lag in the number of practicing nurses [15].

Despite rapid economic growth and increases in expenditures aimed to improve health care, many key health indicators remain low. Although many health status measures show Kazakhstan to be ahead of most of the other nations in the region, it continues to lag behind nations in terms of the size of its economy on several important health and environmental indicators [15]. The service delivery structure remains hospital-oriented and is inadequate to provide high-quality services everywhere. The number of hospital beds has decreased, and the improvement of the hospital sector is not systematic. Despite the rapid growth in the number of PHC specialists, there are too few health workers to ensure equal access to PHC for all. Public health care, long-term care, and rehabilitation are underdeveloped and at all levels of the system, and this is especially true outside large cities, where the provision of services is hampered by poor infrastructure and inadequate equipment. Finally, where data are available, the efficiency and quality of service delivery continues to be significantly lower than it is in most OECD countries. For example, hospitalization rates for outpatient-dependent diseases such as asthma and diabetes are extremely high, and cancer survival rates lag behind.

### Implications for Public Health Policy and Practice

The Republic of Kazakhstan outlined the main directions for the health care system in its *State Program for Healthcare Development 2020–2025*. These include people’s choice in favor of health, a modern public health service, comprehensive health maintenance at the PHC level, the development of human capital, improvement of medical care, the creation of a single digital health care space, the implementation of compulsory health insurance and support for voluntary medical insurance to achieve universal health coverage, improvement in the investment climate in the medical industry, and effective governance in health care [7,27]. Table 1 shows expected changes in the main target indicators.

In order to support health promoting behaviors, there ae planned measured that are intended to increase the health literacy of the population. Projects will be implemented in the media to promote a healthy lifestyle and discourage risky behaviors. The experience of involving non-governmental organizations in public health issues will also be continued and strengthened. Within the framework of intersectoral interaction and on the basis of the WHO recommendation, “Health in All Policies” will be implemented in the regions of the country through the “Healthy Cities and Regions”, including “Health Promoting Schools and Healthy Universities” [28].

Equitable funding for PHC organizations needs to continue to improve if the early detection of disease is to improve in Kazakhstan. The number of visits to the PHC organization per inhabitant per year will thus be monitored and used as an indicator of population use of services aimed at the prevention and early detection of disease.

Diagnosing the root causes of poor health system performance requires not only a high-level assessment of intermediate outcomes and key results but also an understanding of the processes and input resources. Modernization of the information system began a decade ago, but progress has been unpredictable and inconsistent. Overall, the information system is not state-of-the-are and lags behind other OECD countries in terms of the effectiveness of using available data to systematically evaluate and benchmark the improvement of health services. In addition, the exchange of information between health facilities at different levels is limited and constitutes significant obstacles to improving the integration and coordination of treatment and prevention activities. In general, the existing system provides an acceptable level of access to medical services, but there are no robust mechanisms for financial protection against the costs of illness [16].

Increased public funding is likely to be needed to accelerate improvement in treatment outcomes and to bridge gaps in key health indicators. For example, an appropriate volume of cost-effective interventions for chronic disease should be available to all who need them. However, any revision of the range of services provided by medical institutions must be accompanied by appropriate funding to ensure the necessary material and human resources to support implementation.

Expansion of the primary health care system may also be required to better ensure the availability of drugs and services for priority diseases and to eliminate the financial barriers to patient access to care. Many OECD countries use the Health Technology Assessment (HTA) methodology. Based on the assessment results, priorities are highlighted when compiling a package of insurance services, which determine the approach to the selection of drugs included in health insurance, as well as in the development of standard treatment protocols. Today, such protocols are increasingly being developed for non-drug treatments, programs, and services. The Ministry of Health should continue to strengthen its capacity for HTA and explore additional opportunities for international cooperation in this area. Additional targeted and effective investments in health care could enable Kazakhstan to achieve health outcomes that are consistent with its level of economic development. However, in order to improve the efficiency and effectiveness of the health care system, Kazakhstan should consider new strategic priorities in the coming years. To make progress, these should include, but not be limited to, the following:Intensify efforts to reduce chronic diseases.Continue the circle of service delivery reforms.Assure the quality of the system at all levels.Reduce inequality.Strengthen the public health system.Ensure the availability of the guaranteed volume of free medical care (GBMP) and expand coverage of additional cost-effective social benefits.Introduce compulsory social health insurance (CSHI) that is aimed at improving efficiency of the system.

## 5. Conclusions

According to our results, Kazakhstan continues to face many public health challenges in terms of life expectancy trends, maternal mortality, and health system and workforce issues. Despite significant investments in public health, Kazakhstan continues to underperform in terms of life expectancy and other health indicators compared to other OECD countries. The Kazakh economy and the scale and nature of the system of its social institutions, including the present health care system, requires a new approach to its organization and management. Its practical implementation as well as new the institutional conditions for the functioning of the health care system require the health care system to be reformed even further, and such reform must be based on the development of new concepts, legal norms, regulatory procedures, and mechanisms for implementing them—in short, a systemic transformation of the entire health care sector is necessary. The implementation of new reforms may, in fact, have less priority than efforts to ensure the implementation of reforms that are already underway at all levels and that are designed to achieve results on the ground in the short term.

## Figures and Tables

**Figure 1 ijerph-18-12235-f001:**
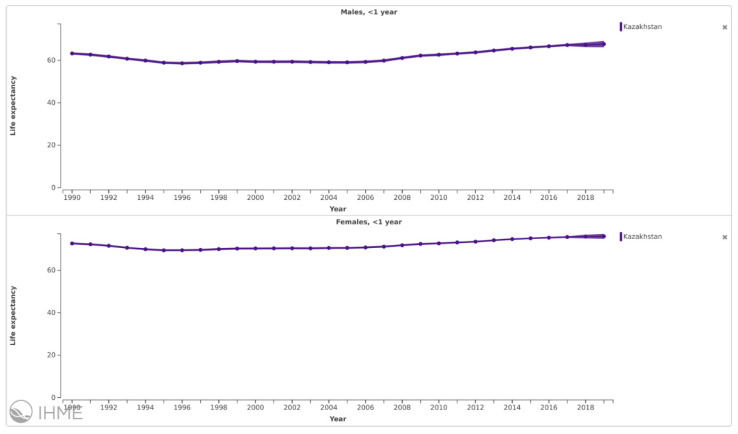
Life expectancy at birth, by gender, 1990 to 2019, Kazakhstan. Source: Institute for Health Metrics and Evaluation. https://vizhub.healthdata.org/gbd-compare/ (accessed on 22 November 2021).

**Figure 2 ijerph-18-12235-f002:**
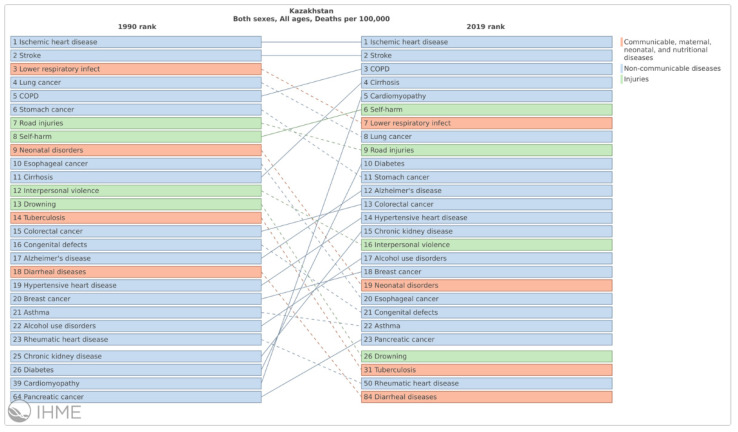
Leading causes of death. Source: Institute for Health Metrics and Evaluation (IHME). https://vizhub.healthdata.org/gbd-compare/ (accessed on 22 November 2021).

**Figure 3 ijerph-18-12235-f003:**
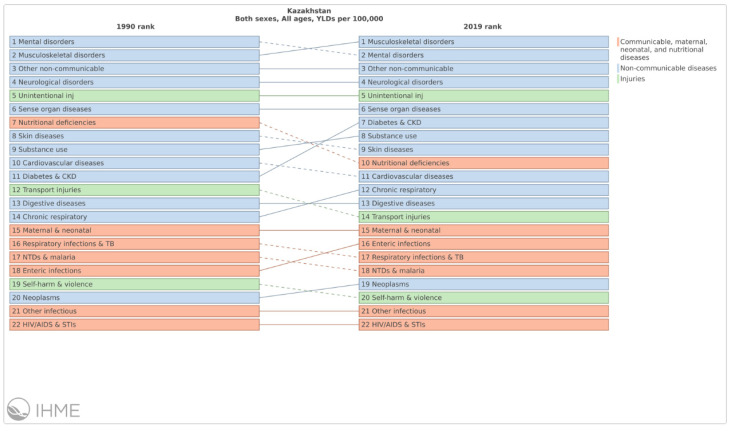
Leading causes of years lived with disability in 1990 and 2019, all ages. Changes signal the increasing impact that obesity and diabetes have on population health. Source: IHME, https://vizhub.healthdata.org/gbd-compare/ (accessed on 22 November 2021).

**Table 1 ijerph-18-12235-t001:** Target indicators for life expectancy and infant and maternal mortality outcomes for the years 2018 to 2025.

Target Indicator	Data Source	2018	2019	2020	2021	2022	2023	2024	2025
Life expectancy at birth	Official statistical information of the Committee on Statistics of the Ministry of National Economy	73.15	73.13	73.21	73.3	73.73	74.15	74.58	75
The level of risk of premature mortality from 30 to 70 years from cardiovascular, cancer, chronic respiratory diseases and diabetes		19.28	19.28	19.28	19.28	19.28	19.28	19.28	19.28
Infant mortality rate, per 1000 life births		10.3	10.3	10.1	9.9	9.6	9.3	8.8	8.3
Maternal mortality rate, per 100,000 live births	Administrative data of the Ministry of Health	17.5	17.4	17.1	16.8	16.3	15.6	15.0	14.5

## Data Availability

The data on which our results are based come from the Global Burden of Disease (GBD) database and toolkit, which are publicly available from the Institute for Health Metrics and Evaluation (IHME) at http://www.healthdata.org/ (accessed on 22 November 2021).

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
