# Peer review of "Population Health Status of the Republic of Kazakhstan: Trends and Implications for Public Health Policy"

_ijerph, 2021, doi:10.3390/ijerph182212235_

Round 1

Reviewer 1 Report

It is really important article for Kazakhstan due to lack of analysis and scientific discussion about many problems in public health of this country.

However, I have some minor comments to authors:

> 1. There is a need to correct punctuation.
> 2. In the abstract, the results should be clarified.
> 3. In the Introduction should be added information with references
> related to hazards of Nuclear test site, because many researches have
> been done.
> 4. Web-links in the Methods section should be referenced in the list of
> references.
> 5. Figure 1 should be corrected, because there is options on the used
> web-link to choose visualization with Confidence intervals. It will give
> a better picture and will add some epidemiological meaning.
> 6. In the discussion section a sentence " Although some of the goals
> have been achieved 210 in some areas, a sharp increase in maternal
> mortality was recorded during the summer of 211 2020 as a consequence of the COVID-19 (SARS-CoV-2) in the country, accounting for 63% 212 of  deaths from the total number of cases. " unclear and should be
referenced. All statements with any results or data should be referenced in the Discussion.
> 7. Conclusion should be based on results of the study.

Author Response

Dear reviewer, 

thank you for your valuable comments! Please see below our responses at your items by color

1. There is a need to correct punctuation. A native English speaker corrected all language and grammar issues
> 2. In the abstract, the results should be clarified We re-wrote the Abstract focusing more on results, however, please be aware we aim to keep it within word limit
> 3. In the Introduction should be added information with references
> related to hazards of Nuclear test site, because many researches have
> been done. We added 2 references
> 4. Web-links in the Methods section should be referenced in the list of
> references. Done, we moved the link to Reference list
> 5. Figure 1 should be corrected, because there is options on the used
> web-link to choose visualization with Confidence intervals. It will give
> a better picture and will add some epidemiological meaning. We replaced the picture with one including uncertainties
> 6. In the discussion section a sentence " Although some of the goals
> have been achieved 210 in some areas, a sharp increase in maternal
> mortality was recorded during the summer of 211 2020 as a consequence of the COVID-19 (SARS-CoV-2) in the country, accounting for 63% 212 of  deaths from the total number of cases. " unclear and should be
referenced. All statements with any results or data should be referenced in the Discussion. We added a reference to that sentence and also to workforce information.
> 7. Conclusion should be based on results of the study We believe it reflects the results now.

Reviewer 2 Report

The Conclusions section is unclear to the reader. The main conclusions of this study are not clearly expressed there. Please rewrite this section. 
For example, you write that Kazakhstan is facing many public health challenges and needs reforms. Please be more specific about both. 

Also, the authors should connect the Conclusions section better with the data in this study. For example "According to GBD data on mortality....".

Author Response

Dear Reviewer,

thank you for your valuable input; please see our responses to you comments directly at your points below.

The Conclusions section is unclear to the reader. The main conclusions of this study are not clearly expressed there. Please rewrite this section. 
For example, you write that Kazakhstan is facing many public health challenges and needs reforms. Please be more specific about both. 

Also, the authors should connect the Conclusions section better with the data in this study. For example "According to GBD data on mortality....". We modified the first sentence in Conclusion to link it more into our findings. However, we want to keep the Conclusion at large extent general, focusing on rather system level and not individual indicator levels therefore we did not repeat the indicator values from Results part. We hope this is understandable and acceptable! 

Reviewer 3 Report

Comments from the Reviewers:

Summary: Overall, this article is promising and oers valuable insight into population health status of the Republic of Kazakhstan_ The article needs a minor revision.

Abstract
Within the abstract, the aim and implications for practice held valuable points for the reader. The abstract was clear. It would be helpful to outline. The conclusions could have benefited to adding your suggestion to improve public health status.

Introduction
This was well laid out. The aim of the study could be clearer: to add what are the current health problems in Kazakhstan and what are the limitations of public health policy.

Materials and Methods

Please describe whether the analysis data Global Burden of Disease (GBD) database and toolkit are freely available to researchers. and briefly add the statements about the health care access and quality index developed by Fullmann [9].

life expectancy, leading cause of mortality, leading 82 causes of disability adjusted life years (DALY’s), risk of premature mortality due to non-83 communicable diseases, infant mortality, and maternal mortality – You need to describe that these variables represents the current status of public health in Kzakhstan. Are there all indicators to represent of public health?

Which program you use to analyses?

Results

Years word is required for the vertical axis in Figure 1

It would be good to briefly fill out the contents of the child mortality in a Table

Discussion
While the discussion relates to the findings and there was application of literature to widen the debate.

The researchers suggest new seven strategies and need to illustrate each strategy in more detail. For example, you could suggest how to reduce inequality? Or which way to strengthen the public health system.

Thank you again for asking me to review this article. I would be happy to review this article again following amendments.

Author Response

Dear Reviewer,

thank you for your valuable comments! Please find our responses below directly at your items by color.

Summary: Overall, this article is promising and oers valuable insight into population health status of the Republic of Kazakhstan_ The article needs a minor revision.

Thank you!
Abstract
Within the abstract, the aim and implications for practice held valuable points for the reader. The abstract was clear. It would be helpful to outline. The conclusions could have benefited to adding your suggestion to improve public health status.

We re-wrote the Abstract and added the seven strategic priorities into last sentence
Introduction
This was well laid out. The aim of the study could be clearer: to add what are the current health problems in Kazakhstan and what are the limitations of public health policy.

We modified the sentence with the aim of the study making it clear that we aim to review the trend of population health using selected indicators and based on it to discuss the public health policy. The limitation of public health policy are discussed in Discussion as we believe, first we need to have the Results and then relate them to the policy.
Materials and Methods

Please describe whether the analysis data Global Burden of Disease (GBD) database and toolkit are freely available to researchers. and briefly add the statements about the health care access and quality index developed by Fullmann [9]. We modified the text making it clear that the GBD database and toolkit is freely available to researchers. In terms of health care access and quality index we remain with standard publishing procedures and keep the reference only, without prescribing the index method. 

life expectancy, leading cause of mortality, leading 82 causes of disability adjusted life years (DALY’s), risk of premature mortality due to non-83 communicable diseases, infant mortality, and maternal mortality – You need to describe that these variables represents the current status of public health in Kzakhstan. Are there all indicators to represent of public health?

No, of course these are not all indicators to assess status of population health. These are however the indicators the author team agreed upon and could get access to data based on public databases. They are very often used in public health literature for such purpose. Other authors could use different indicators of course.

Which program you use to analyses?

As mentioned in methods, we used directly the GBD toolkit.
Results

Years word is required for the vertical axis in Figure 1 we replaced Figure 1 by new figure including uncertainties and also the axis label

It would be good to briefly fill out the contents of the child mortality in a Table

We are not sure what did you mean by this comment! The name of indicator, source of data as well as the expected target values are there, we have nothing more to add there.

Discussion
While the discussion relates to the findings and there was application of literature to widen the debate. Thank you!

The researchers suggest new seven strategies and need to illustrate each strategy in more detail. For example, you could suggest how to reduce inequality? Or which way to strengthen the public health system.

We believe what you are asking for would require a new manuscript! Such manuscript should focus on different issues from Introduction, through results and final Discussion and Conclusions. It would also require partially different author team to include Kazakh policy makers of different sectors; only that would allow us to describe methods to reduce inequalities! So, this is a request, which goes beyond the aim of our manuscript and we are sorry for not being able to address it in this case.